# Antimicrobial Resistance in *Salmonella* Isolated from Food Workers and Chicken Products in Japan

**DOI:** 10.3390/antibiotics10121541

**Published:** 2021-12-16

**Authors:** Yoshimasa Sasaki, Hiromi Kakizawa, Youichi Baba, Takeshi Ito, Yukari Haremaki, Masaru Yonemichi, Tetsuya Ikeda, Makoto Kuroda, Kenji Ohya, Yukiko Hara-Kudo, Tetsuo Asai, Hiroshi Asakura

**Affiliations:** 1Division of Biomedical Food Research, National Institute of Health Sciences, 3-25-26, Tonomachi, Kawasaki-ku, Kawasaki 210-9501, Kanagawa, Japan; hasakura@nihs.go.jp; 2Department of Applied Veterinary Science, The United Graduate School of Veterinary Science, Gifu University, 1-1, Yanagido, Gifu 501-1193, Gifu, Japan; tasai@gifu-u.ac.jp; 3Incorporated Foundation Tokyo Kenbikyo-in, 1-100-38 Takamatsu-cho, Tachikawa 190-0011, Tokyo, Japan; kakizawa@kenko-kenbi.or.jp (H.K.); baba@kenko-kenbi.or.jp (Y.B.); tito@kenko-kenbi.or.jp (T.I.); 4BML Food Science Solutions, Inc., 1549-7, Matoba, Kawagoe 350-1101, Saitama, Japan; yharemak@bml.co.jp (Y.H.); m-yonemi@bml.co.jp (M.Y.); 5Department of Infectious Diseases, Hokkaido Institute of Public Health, Kita19 Nishi 12, Kita-ku, Sapporo 060-0819, Hokkaido, Japan; ikeda@iph.pref.hokkaido.jp; 6Pathogen Genomics Center, National Institute of Infectious Diseases, 1-23-1 Toyama, Shinjuku-ku, Tokyo 162-8640, Japan; makokuro@niid.go.jp; 7Division of Microbiology, National Institute of Health Sciences, 3-25-26, Tonomachi, Kawasaki-ku, Kawasaki 210-9501, Kanagawa, Japan; kohya@nihs.go.jp (K.O.); ykudo@nihs.go.jp (Y.H.-K.)

**Keywords:** antimicrobial resistance, *Salmonella*, food worker, chicken product

## Abstract

*Salmonella* is an enteric bacterial pathogen that causes foodborne illness in humans. Third-generation cephalosporin (TGC) resistance in *Salmonella* remains a global concern. Food workers may represent a reservoir of *Salmonella*, thus potentially contaminating food products. Therefore, we aimed to investigate the prevalence of *Salmonella* in food workers and characterize the isolates by serotyping and antimicrobial susceptibility testing. *Salmonella* was isolated from 583 (0.079%) of 740,635 stool samples collected from food workers between January and December 2018, and then serotyped into 76 *Salmonella enterica* serovars and 22 untypeable *Salmonella* strains. High rates of antimicrobial resistance were observed for streptomycin (51.1%), tetracycline (33.1%), and kanamycin (18.4%). Although isolates were susceptible to ciprofloxacin, 12 (2.1%) strains (one *S*. Infantis, one *S*. Manhattan, two *S*. Bareilly, two *S*. Blockley, two *S*. Heidelberg, two *S*. Minnesota, one *S*. Goldcoast, and one untypeable *Salmonella* strain) were resistant to the TGC cefotaxime, all of which harbored β-lactamase genes (*bla*_CMY-2_, *bla*_CTX-M-15_, *bla*_CTX-M-55_, and *bla*_TEM-52B_). Moreover, 1.3% (4/309) of *Salmonella* strains (three *S*. Infantis and one *S*. Manhattan strains) isolated from chicken products were resistant to cefotaxime and harbored *bla*_CMY-2_ or *bla*_TEM-52B_. Thus, food workers may acquire TGC-resistant *Salmonella* after the ingestion of contaminated chicken products and further contaminate food products.

## 1. Introduction

Non-typhoidal *Salmonella* is an enteric bacterial pathogen that causes foodborne illnesses worldwide. Salmonellosis is caused by non-typhoidal *Salmonella enterica* serovars such as *Salmonella enterica* subsp. *enterica* serovar Enteritidis, *Salmonella* Typhimurium, and *Salmonella* Infantis. Invasive non-typhoidal *Salmonella* is estimated to cause 3.4 million cases of infections annually, worldwide [1]. Although non-typhoidal *Salmonella* infections typically cause acute self-limiting gastroenteritis, they may cause invasive bacteremia, with severe symptoms in certain cases. Antimicrobial therapy is typically not provided in many cases of salmonellosis. However, antimicrobial therapy may be lifesaving in patients with bacteremia or an extra-intestinal focal infection. Ampicillin, third-generation cephalosporins (TGCs), fluoroquinolones, and trimethoprim-sulfamethoxazole are commonly used to treat salmonellosis [2]. Therefore, antimicrobial-resistant *Salmonella* poses an important issue in the chemotherapy treatment of humans.

A primary route of *Salmonella* transmission involves ingestion of contaminated chicken meat and eggs [3]. Mori et al. [4] isolated *Salmonella* from 286 (55.9%) of 512 chicken meat samples collected in Japan between June 2015 and January 2016. In their study, the most frequently isolated serovar was *S.* Infantis, followed by *S.* Schwarzengrund, together accounting for 78.2% (243/311) of the isolates; 74.9% (182/243) of these two serovars are resistant to two or more antimicrobials, and seven *S*. Infantis strains were resistant to cefotaxime, a TGC. TGC-resistant *Salmonella* has additionally been isolated from fecal samples from healthy adults engaged in food handling work and diarrheic patients in Japan between 2012 and 2015 [5,6]. TGCs are classified as “critically important” by the World Health Organization [7]. In Japan, ceftiofur, a TGC, was approved in 1996 and was used for chemotherapy in cattle and pigs with bacterial infections. Moreover, the off-label use of ceftiofur in combination with *in-ovo* vaccination or vaccination of newly hatched chicks was performed in certain hatcheries until its use was abandoned in March 2012 [8]. The recent isolation of TGC-resistant *Salmonella* from chicken meat revealed TGC-resistant *Salmonella* may have survived in hatcheries and chicken farms after the withdrawal of ceftiofur in hatcheries. Food workers including food handlers who work in food and beverage companies that are infected with antimicrobial-resistant *Salmonella* after consuming or handling contaminated food may serve as reservoirs and pose a risk for food contamination. Therefore, in this study, we aimed to clarify the prevalence of *Salmonella* in food workers and characterize the isolates through serotyping and antimicrobial susceptibility testing. To estimate the origin of the isolates, we characterized *Salmonella* strains isolated from local chicken products by serotyping and antimicrobial susceptibility testing, and then compared the characteristics between human- and chicken-derived strains. Based on the findings of this study, the prevalence of *Salmonella* in stool samples of food workers was 0.079% between January and December 2018. Additionally, food workers may represent a reservoir of TGC-resistant *Salmonella*.

## 2. Results

### 2.1. Salmonella Prevalence in Human Stools and Local Chicken Products

*Salmonella* was isolated from 583 (0.079%, 95% confidence interval = 0.072–0.085) of 740,635 human stool samples. The monthly prevalence of *Salmonella* ranged from 0.050% (32/63,727) in January to 0.0125% (78/62,417) in September. The prevalence of *Salmonella* in human stool samples was significantly greater (0.101%; 188/185,718; Fisher’s exact test, *p* < 0.01) from August to October than from January to March (0.067%; 125/186,208). *Salmonella*-positive samples were obtained from 45 of 47 prefectures in Japan. The strains isolated from humans were serotyped into 76 serovars and 22 untypeable *Salmonella* isolates (Table 1). The top 14 serovars (*S*. Schwarzengrund to *S*. Typhimurium) together accounted for over 70% of human-derived strains. Of 235 local chicken products, *Salmonella* was isolated from 200 (85.1%, 95% confidence interval = 79.9–89.4). To estimate the origin of the isolates, we characterized *Salmonella* strains isolated from local chicken products. Chicken-derived strains were serotyped into five serovars and two untypeable *Salmonella* strains. The top three serovars were *S*. Schwarzengrund (73.0%), *S*. Infantis (15.0%), and *S.* Manhattan (8.5%), together accounting for 96.5% of the chicken-derived strains.

### 2.2. Antimicrobial Susceptibility

Antimicrobial susceptibility testing of human-derived strains showed that 333 (57.1%) of the 583 strains were resistant to at least one antimicrobial, and 204 (35.0%) isolates were resistant to two or more antimicrobials. High rates of antimicrobial resistance were observed for streptomycin (51.1%), tetracycline (33.1%), and kanamycin (18.4%) (Table 2). All isolates were susceptible to ciprofloxacin. In total, (2.1%, 12/583) 12 strains (one *S.* Infantis, one *S*. Manhattan, two *S.* Bareilly, two *S.* Blockley, two *S.* Heidelberg, two *S.* Minnesota, one *S.* Goldcoast, and one untypeable *Salmonella* strain) were resistant to cefotaxime. The cefotaxime-resistant untypeable strain agglutinated with anti-H:r, anti-H:1, and anti-H:5 sera, but not with anti-O sera. Multidrug-resistant strains isolated from human stool samples such as *S*. Schwarzengrund, *S*. Infantis, *S*. Typhimurium monophasic variant, and *S*. Manhattan accounted for 86.9%, 28.0%, 75.7%, and 88.0%, respectively, together accounting for 73.5% (150/204) (Table 3). Among *S.* Schwarzengrund strains, the prevalence of strains resistant to streptomycin, kanamycin, and tetracycline was greatest (58.6%), followed by streptomycin and tetracycline (11.1%). Among *S*. Infantis strains, the prevalence of strains susceptible to antimicrobials tested was greatest (40.0%), followed by streptomycin (32.0%). Among *S*. Typhimurium monophasic variants, the prevalence of strains resistant to ampicillin, streptomycin, and tetracycline was greatest (45.9%). Among *S*. Manhattan strains, the prevalence of strains resistant to streptomycin and tetracycline was greatest (60.0%). In contrast, multidrug-resistant strains were not observed among *S.* Thompson, *S.* Cubana, and *S.* Newport strains.

Antimicrobial susceptibility testing of chicken-derived strains showed that 174 (87.0%) of the 200 strains were resistant to at least one antimicrobial, and 141 (70.5%) strains were resistant to two or more antimicrobials. High antimicrobial resistance rates were observed for streptomycin (73.0%), tetracycline (67.5%), and kanamycin (53.5%). Four (1.3%) strains (three *S*. Infantis and one *S*. Manhattan strain) were resistant to cefotaxime. The three products contaminated with these cefotaxime-resistant *S*. Infantis strains were transported from the same chicken-processing plant on different days. The product contaminated with cefotaxime-resistant *S.* Manhattan was transferred from a different chicken processing plant. Among *S*. Schwarzengrund strains, the prevalence of strains resistant to streptomycin, kanamycin, and tetracycline was greatest (32.2%), sharing similarity with human-derived strains. Among *S*. Manhattan strains, the prevalence of strains resistant to streptomycin and tetracycline was greatest (50.0%), bearing similarity with human-derived strains. Among *S*. Infantis strains, the prevalence of strains resistant to streptomycin and tetracycline was greatest (36.7%), which differed from that in human-derived strains. All strains were susceptible to colistin and ciprofloxacin.

Whole-genome sequencing (WGS) analysis revealed that all 16 cefotaxime-resistant strains isolated from human stool samples and chicken products harbored β-lactamase (*bla*) genes (Table 4). Of the 16 strains, 8 (4 *S.* Infantis, 2 *S*. Minnesota, 1 *S*. Heidelberg, and 1 untypeable *Salmonella* strain) harbored *bla*_CMY-2_. Two *S.* Blockley strains harbored *bla*_CTX-M-15_. One *S.* Heidelberg strain harbored *bla*_LAT-3_. Two *S.* Bareilly strains harbored *bla*_CTX-M-55_. Two *S*. Manhattan strains harbored *bla*_TEM-52B_, and one *S*. Goldcoast strain harbored *bla*_CTX-M-55_, *bla*_LAP-2,_ and *bla*_TEM-1_.

## 3. Discussion

The results of the present study showed that the prevalence of *Salmonella* in stool samples of food workers was 0.079% between January and December 2018, and the prevalence was greater from August to October than that from January to March. Outbreaks of *Salmonella*-associated food poisoning in Japan occur year-round, peaking from May to November [3]. Xu et al. [9] have suggested *Salmonella* may cause human infections in symptomatic and asymptomatic states, as *Salmonella* strains isolated from symptomatic and asymptomatic individuals are genetically and phenotypically indistinguishable. Kariuki et al. [10] reported an elevated level of relatedness between *Salmonella* strains isolated from symptomatic patients and asymptomatic carriers by phylogenetic analysis. Food workers evaluated in the present study may have been infected with *Salmonella* via ingestion of foods that can cause food poisoning, although they were asymptomatic when submitting their stool samples. Excluding *S*. Typhimurium monophasic variants, *S*. Cubana, and *S*. Newport, the top 14 serovars in human-derived strains, are known to be prevalent in Japanese broiler and laying hen flocks [11,12]. Notably, three serovars, *S*. Schwarzengrund, *S*. Infantis, and *S*. Manhattan, accounted for over 98% of chicken-derived strains in the present study. The most common antimicrobial resistance profiles of *S*. Schwarzengrund and *S*. Manhattan in human-derived strains were the same as those in chicken-derived strains. However, the primary antimicrobial resistance profiles of *S*. Infantis in human- and chicken-derived strains were different. *S*. Infantis strains susceptible to all antimicrobials tested represented the most common isolates among human-derived strains, whereas they represented the second-most common strains among chicken-derived strains. In our previous study [11], *S*. Infantis represented the second-most common serovar among *Salmonella* strains isolated from Japanese laying hen flocks, with 92.9% of *S*. Infantis strains susceptible to all antimicrobials tested. Therefore, the distribution of antimicrobial resistance profiles in *S*. Infantis isolated from human stool samples in the present study may be attributed to strains originating from Japanese broilers and laying hen flocks. Shimojima et al. [13] recently reported that *Salmonella* prevalence was greater in local chicken products (57.9%) than in imported products (8.5%). According to the Ministry of Agriculture, Forestry and Fisheries of Japan [14], local chicken and chicken egg products accounted for approximately 65% of total chicken meat distribution and 96% of total chicken egg distribution, respectively, in Japan in FY 2018. The results of this study suggest many food workers may be infected with *Salmonella* via ingestion of local chicken and egg products. Moreover, as *S*. Typhimurium, its monophasic variant, and *S*. Newport are prevalent in local cattle and pigs [15], beef and pork may represent sources of *Salmonella* infections among food workers.

Of the 583 human-derived *Salmonella* strains isolated in the present study, 12 (2.1%) were resistant to cefotaxime. Cefotaxime resistance was observed in seven serovars and one untypeable *Salmonella* strain. As the untypeable strain was assigned to sequence type 32, to which most *S*. Infantis strains belong, the strain may be a variant of *S*. Infantis. This assumption is based on multi-locus sequence typing (MLST) in accordance with public databases for molecular typing and microbial genome diversity (https://pubmlst.org). In the present study, cefotaxime-resistant *S*. Infantis and *S*. Manhattan strains were obtained from local chicken products, and harbored the same *bla* genes (*bla*_CMY-2_ or *bla*_TEM-52B_) as strains isolated from human stool samples. Shigemura et al. [16] reported that *bla*_CMY-2_-harboring *S*. Infantis and *bla*_TEM-52_-harboring *S*. Manhattan strains were isolated from local chicken products, and one *bla*_CMY-2_-harboring *S*. Heidelberg strain was isolated from an imported chicken product. According to the Trade Statistics of Japan (https://www.customs.go.jp/toukei/info/index_e.htm), chicken products imported from Brazil accounted for approximately 75% of imported chicken products in 2018. In Brazil, *S*. Minnesota, *S*. Infantis, and *S*. Heidelberg are prevalent in broiler flocks and chicken products, some of which show TGC resistance [17,18,19,20,21]. Shimojima et al. [13] reported the cefotaxime resistance rate in *Salmonella* isolated between 2009 and 2017 was lower in local chicken products (1.9%) than in imported chicken products (28.0%); additionally, seven *S*. Heidelberg and three *S*. Minnesota strains, isolated from imported chicken products, contain *bla* genes. Moreover, in their study, *bla*-harboring *Salmonella* strains were not isolated from beef and pork samples. Therefore, local and imported chicken products may represent major sources of TGC-resistant *Salmonella* that infect food workers.

Shigemura et al. [22] recently reported the prevalence of *Salmonella* in human stool samples obtained from food workers between January and October 2017 was 0.113% (164/145,220), and seven strains were resistant to TGCs. Of the seven TGC-resistant strains, one *Salmonella* Senftenberg and three *Salmonella* Haardt strains harbored *bla*_CTX-M-14_ and *bla*_CTX-M-15_, respectively. The remaining three strains, namely *Salmonella* Agona, *S*. Infantis, and an untypeable *Salmonella* strain, harbored *bla*_CMY-2_. Although their detection method differed from ours, their results share similarity to those reported in the present study, suggesting food workers may be infected with TGC-resistant *Salmonella* after the ingestion of contaminated chicken products and subsequently serve as reservoirs. Although the prevalence of TGC-resistant *Salmonella* in food workers may be low, they routinely serve food to consumers. Moreover, estimating the prevalence and characteristics of *Salmonella* in the Japanese population including food workers is possible. Therefore, monitoring the prevalence and characteristics of *Salmonella* in food workers is useful for risk management of *Salmonella*-associated food poisoning.

## 4. Materials and Methods

### 4.1. Sample Collection and Salmonella Isolation

A total of 740,635 stool samples (approximately 60,000 samples per month) were collected from food workers in all 47 prefectures in Japan between January and December 2018. The food workers included cooks and servers in restaurants and food factory workers. The individuals periodically (mostly monthly) collected their stool samples using transport swabs with a modified Cary-Blair transport medium (Eiken Chimical Co., Tokyo, Japan) and then submitted them to one of two laboratories (the Incorporated Foundation Tokyo Kenbikyo-in and the BML Food Science Solutions) during the study period. We were not provided with the precise number of individuals who submitted stool samples or the number of facilities involved in sampling. The individuals did not report symptoms of enteritis upon submission of stool samples. Each sample was tested in laboratories within 72 h upon arrival. A tip of each swab was streaked onto a modified Salmonella–Shigella agar plate (Eiken Chemical) and incubated at 37 °C for 24 h. Putative *Salmonella* colonies were biochemically identified as described previously [23]. One strain per sample was suspended in 20% glycerol and stored at −80 °C until ready for serotyping and antimicrobial susceptibility testing.

### 4.2. Local Chicken Products and Salmonella Isolation

A total of 235 local chicken products (liver, breast, thigh, neck skin, and minced meat) were collected from 25 retail stores and 6 chicken processing plants between January 2018 and October 2021. The products were transported under refrigeration to the National Institute of Health Sciences. Each sample was tested within 24 h after arrival at the laboratory. Twenty-five grams of the sample was mixed in 225 mL of buffered peptone water (Oxoid Ltd., Hampshire, UK) and incubated at 37 °C for 18 h for pre-enrichment. After incubation, 0.1 and 1 mL of the culture was added to 10 mL of Rappaport–Vassiliadis broth (Oxoid) and 10 mL of Hajna tetrathionate broth (Eiken Chemical), respectively, then incubated at 42 °C for 20 h. After incubation, each culture was streaked onto two selective isolation agar plates: xylose-lysine-deoxycholate agar (Oxoid) and CHROMagar^TM^ Salmonella (CHROMagar, Paris, France) then incubated at 37 °C for 24 h. For the selective isolation from minced meat, Mannitol-Lysine-Crystal-Violet-Brilliant-Green agar (Nissui Pharmaceutical, Tokyo, Japan) was used in place of xylose-lysine-deoxycholate agar. Putative *Salmonella* colonies were biochemically identified as mentioned above. One strain per sample was suspended in 20% glycerol and stored at −80 °C until ready for serotyping and antimicrobial susceptibility testing.

### 4.3. Serotyping

*Salmonella* strains were tested for somatic antigens by slide agglutination using O antisera (Denka Co., Tokyo, Japan, and SSI Diagnostica, Copenhagen, Denmark). *Salmonella* strains were further tested for flagella antigens by tube agglutination using H antisera (Denka). Serovars were determined based on the reaction between O and H group antigens according to the Kauffmann–White scheme [24]. Strains agglutinated with anti-O4 and anti-H:i serum but not anti-H:1 or anti-H:2 serum were confirmed as monophasic variants of *S*. Typhimurium using a previously reported polymerase chain reaction method [25].

### 4.4. Antimicrobial Susceptibility Testing

The susceptibility of the *Salmonella* isolates to 11 antimicrobial agents was determined by antimicrobial susceptibility testing. Minimum inhibitory concentrations were determined by a two-fold broth microdilution method in 96-well microtiter plates (Dry-plate ‘Eiken’, Eiken Chemical), as described in the standards of the Clinical and Laboratory Standards Institute (CLSI) [26]. The following antimicrobial agents were tested: ampicillin (range of antimicrobial dilution: 1 to 128 mg/L), cefazolin (1 to 128 mg/L), cefotaxime (0.5 to 64 mg/L), streptomycin (1 to 128 mg/L), gentamicin (0.5 to 64 mg/L), kanamycin (1 to 128 mg/L), tetracycline (0.5 to 64 mg/L), nalidixic acid (1 to 128 mg/L), ciprofloxacin (0.03 to 4 mg/L), colistin (0.12 to 16 mg/L), and chloramphenicol (1 to 128 mg/L). The *Escherichia coli* reference strain ATCC 25922 was used as a control. The plates were incubated for 24 ± 2 h at 37 °C under aerobic conditions. The resistance breakpoints were defined based on the CLSI standard [27] and the Report on the Japanese Veterinary Antimicrobial Resistance Monitoring System 2016–2017 [28].

### 4.5. Determination of Antimicrobial Resistance Genes and Sequence Types Based on MLST in Cefotaxime-Resistant Salmonella Strains by WGS Analysis

DNA was extracted from cefotaxime-resistant strains using DNeasy^Ⓡ^ UltraClean^Ⓡ^ Microbial Kit (Qiagen GmbH, Hilden, Germany). WGS analysis was performed as previously described [29]. Sequencing libraries for each strain were prepared using QIAseq FX Library Kit (Qiagen) to obtain pair-end sequences (300 bp × 2) using the Illumina Miseq platform. Draft genome sequences were obtained by assembling the read sequences using A5 miseq [30]. The WGS data were analyzed using the Database of Pathogen Genomics and Epidemiology, GenEpid-J [31], and by using the ResFinder tool available from the Center for Genomic Epidemiology (http://www.genomicepidemiology.org/, accessed on 21 July 2021). MLST was performed using nucleotide sequences of seven housekeeping genes (*aroC*, *dnaN*, *hemD*, *hisD*, *purE*, *sucA*, and *thrA*) according to protocols available on the MLST database (https://pubmlst.org/organisms/salmonella-spp/, accessed on 23 November 2021).

### 4.6. Statistical Analysis

All statistical analyses were performed using R version 4.1. Confidence intervals were determined using the exact binomial test. Differences between proportions were tested using Fisher’s exact test, where *p*-values of <0.05 were considered statistically significant.

## 5. Conclusions

*Salmonella* was isolated from 0.079% (583/740,635) of stool samples obtained from food workers in Japan. The strains were serotyped into 76 serovars and 22 untypeable *Salmonella* isolates. The top 14 serovars together accounted for over 70% of human-derived strains. A majority of the top 14 serovars in human-derived strains are prevalent in Japanese broiler and laying hen flocks. Of the 583 *Salmonella* strains, 12 (2.1%) were resistant to the TGC cefotaxime and harbored the same *bla* genes as those observed in local and imported chicken products. The characterization of *Salmonella* isolated from food workers showed that local and imported chicken products may represent the primary sources of TGC-resistant *Salmonella* responsible for infection among food workers. Although the prevalence of TGC-resistant *Salmonella* in food workers is low, they may contaminate food served to consumers. Therefore, monitoring the prevalence and characteristics of *Salmonella* in food workers is necessary for the risk management of *Salmonella*-associated food poisoning.

## Figures and Tables

**Table 1 antibiotics-10-01541-t001:** Number and serovars of *Salmonella* strains isolated from humans and chicken products.

Serovar	Human		Chicken	
	N	%	N	%
Schwarzengrund	99	17.0	146	73.0
Infantis	50	8.6	30	15.0
Typhimurium monophasic variant	37	6.3	0	0.0
Thompson	36	6.2	0	0.0
Cubana	26	4.5	0	0.0
Manhattan	25	4.3	17	8.5
Newport	25	4.3	0	0.0
Mbandaka	20	3.4	0	0.0
Agona	17	2.9	4	2.0
Bareilly	16	2.7	0	0.0
Braenderup	16	2.7	0	0.0
Corvallis	16	2.7	0	0.0
Enteritidis	14	2.4	0	0.0
Typhimurium	13	2.2	0	0.0
Saintpaul	9	1.5	0	0.0
Anatum	8	1.4	1	0.5
Stanley	7	1.2	0	0.0
Litchfield	6	1.0	0	0.0
Rissen	6	1.0	0	0.0
Senftenberg	6	1.0	0	0.0
Blockley	5	0.9	0	0.0
Weltevreden	5	0.9	0	0.0
Heidelberg	5	0.9	0	0.0
Derby	4	0.7	0	0.0
Hadar	4	0.7	0	0.0
Panama	4	0.7	0	0.0
Paratyphi B	4	0.7	0	0.0
Colindale	3	0.5	0	0.0
London	3	0.5	0	0.0
Montevideo	3	0.5	0	0.0
Muenchen	3	0.5	0	0.0
Oranienburg	3	0.5	0	0.0
Othmarschen	3	0.5	0	0.0
Potsdam	3	0.5	0	0.0
Uganda	3	0.5	0	0.0
Altona	2	0.3	0	0.0
Brandenburg	2	0.3	0	0.0
Bredeney	2	0.3	0	0.0
Cerro	2	0.3	0	0.0
Duesseldorf	2	0.3	0	0.0
Havana	2	0.3	0	0.0
Lexington	2	0.3	0	0.0
Liverpool	2	0.3	0	0.0
Minnesota	2	0.3	0	0.0
Narashino	2	0.3	0	0.0
Poona	2	0.3	0	0.0
Singapore	2	0.3	0	0.0
Virchow	2	0.3	0	0.0
Others (28 serovars and 22 untypeable)	50	8.6	2	1.0
Total	583		200	

**Table 2 antibiotics-10-01541-t002:** Antimicrobial susceptibility of Salmonella isolates from humans and chicken products.

Serovar	Origin	N	ABPC		CEZ		CTX		SM		GM		KM		TC		NA		CL		CP	
			N	%	N	%	N	%	N	%	N	%	N	%	N	%	N	%	N	%	N	%
Schwarzengrund	Human	99	0	0.0	0	0.0	0	0.0	85	85.9	0	0.0	81	81.8	83	83.8	16	16.2	0	0.0	0	0.0
	Chicken	146	1	0.7	0	0.0	0	0.0	99	67.8	0	0.0	93	63.7	93	63.7	24	16.4	0	0.0	2	1.4
Infantis	Human	50	1	2.0	1	2.0	1	2.0	30	60.0	0	0.0	8	16.0	13	26.0	1	2.0	0	0.0	0	0.0
	Chiciken	30	4	13.3	3	10.0	3	10.0	26	86.7	1	3.3	10	33.3	24	80.0	4	13.3	0	0.0	0	0.0
Typhimurium monophasic variant	Human	37	21	56.8	1	2.7	0	0.0	29	78.4	1	2.7	1	2.7	29	78.4	1	2.7	0	0.0	2	5.4
Thompson	Human	36	0	0.0	0	0.0	0	0.0	21	58.3	0	0.0	0	0.0	0	0.0	0	0.0	0	0.0	0	0.0
Cubana	Human	26	0	0.0	0	0.0	0	0.0	1	3.8	0	0.0	0	0.0	0	0.0	0	0.0	0	0.0	0	0.0
Manhattan	Human	25	5	20.0	1	4.0	1	4.0	24	96.0	0	0.0	0	0.0	22	88.0	6	24.0	0	0.0	0	0.0
	Chicken	17	1	5.9	1	5.9	1	5.9	15	88.2	0	0.0	1	5.9	12	70.6	1	5.9	0	0.0	0	0.0
Newport	Human	25	0	0.0	0	0.0	0	0.0	3	12.0	0	0.0	0	0.0	0	0.0	0	0.0	0	0.0	0	0.0
Mbandaka	Human	20	0	0.0	0	0.0	0	0.0	8	40.0	0	0.0	0	0.0	1	5.0	0	0.0	0	0.0	0	0.0
Agona	Human	17	1	5.9	1	5.9	0	0.0	9	52.9	0	0.0	0	0.0	10	58.8	2	11.8	0	0.0	1	5.9
	Chicken	4	0	0.0	0	0.0	0	0.0	4	100.0	0	0.0	1	25.0	4	100.0	0	0.0	0	0.0	1	25.0
Bareilly	Human	16	2	12.5	2	12.5	2	12.5	5	31.3	2	12.5	0	0.0	0	0.0	0	0.0	0	0.0	0	0.0
Braenderup	Human	16	0	0.0	0	0.0	0	0.0	12	75.0	0	0.0	0	0.0	0	0.0	2	12.5	0	0.0	0	0.0
Corvallis	Human	16	0	0.0	0	0.0	0	0.0	1	6.3	0	0.0	0	0.0	0	0.0	0	0.0	0	0.0	0	0.0
Enteritidis	Human	14	1	7.1	0	0.0	0	0.0	1	7.1	0	0.0	0	0.0	0	0.0	2	14.3	2	14.3	0	0.0
Typhimurium	Human	13	0	0.0	0	0.0	0	0.0	2	15.4	0	0.0	0	0.0	1	7.7	0	0.0	0	0.0	0	0.0
Others	Human	173	20	11.6	9	5.2	8	4.6	67	38.7	2	1.2	17	9.8	34	19.7	11	6.4	0	0.0	10	5.8
	Chicken	3	0	0.0	0	0.0	0	0.0	2	66.7	0	0.0	2	66.7	2	66.7	0	0.0	0	0.0	0	0.0
Total	Human	583	51	8.7	15	2.6	12	2.1	298	51.1	5	0.9	107	18.4	193	33.1	41	7.0	2	0.3	13	2.2
	Chicken	200	6	3.0	4	2.0	4	2.0	146	73.0	1	0.5	107	53.5	135	67.5	29	14.5	0	0.0	3	1.5

ABPC: ampicillin, CEZ: cefazolin, CTX: cefotaxime, SM: streptomycin, GM: gentamycin, KM: kanamycin, TC: tetracycline, NA: nalidixic acid, CL: colistin, CP: chloramphenicol.

**Table 3 antibiotics-10-01541-t003:** Antimicrobial resistance profiles of the six most-frequent *Salmonella enterica* serovars isolated from human stools and chicken products.

Serovar	Antimicrobial Resistance Profile	Human		Chicken	
		No.	%	No.	%
Schwarzengrund		99		146	
	ABPC+SM+KM+TC+NA+CP	0	0.0	1	0.7
	SM+KM+TC+NA+CP	0	0.0	1	0.7
	SM+KM+TC+NA	8	8.1	16	11.0
	KM+TC+NA	1	1.0	0	0.0
	SM+KM+NA	1	1.0	1	0.7
	SM+TC+NA	5	5.1	3	2.1
	SM+KM+TC	58	58.6	47	32.2
	SM+TC	11	11.1	24	16.4
	SM+KM	1	1.0	3	2.1
	KM+NA	1	1.0	2	1.4
	KM+TC	0	0.0	1	0.7
	KM	11	11.1	21	14.4
	SM	1	1.0	3	2.1
	susceptible	1	1.0	23	15.8
Infantis		50		30	
	ABPC+SM+GM+KM+TC+NA	0	0.0	1	3.3
	ABPC+CEZ+CTX+SM+KM+TC	1	2.0	0	0.0
	ABPC+CEZ+CTX+SM+TC	0	0.0	3	10.0
	SM+KM+TC+NA	0	0.0	1	3.3
	SM+KM+TC	6	12.0	6	20.0
	SM+TC+NA	1	2.0	2	6.7
	SM+TC	5	10.0	11	36.7
	SM+KM	1	2.0	0	0.0
	TC	0	0.0	0	0.0
	KM	0	0.0	2	6.7
	SM	16	32.0	2	6.7
	susceptible	20	40.0	2	6.7
Typhimurium monophasic variant		37		0	
	SM+GM+KM+TC	1	2.7	0	0.0
	ABPC+CEZ+SM+TC	1	2.7	0	0.0
	ABPC+SM+TC+CP	1	2.7	0	0.0
	ABPC+SM+NA	1	2.7	0	0.0
	SM+TC+CP	1	2.7	0	0.0
	ABPC+SM+TC	17	45.9	0	0.0
	ABPC+SM	1	2.7	0	0.0
	SM+TC	5	13.5	0	0.0
	TC	3	8.1	0	0.0
	SM	1	2.7	0	0.0
	susceptible	5	13.5	0	0.0
Thompson		36		0	
	SM	21	58.3	0	0.0
	susceptible	15	41.7	0	0.0
Cubana		26		0	
	SM	1	3.8	0	0.0
	susceptible	25	96.2	0	0.0
Manhattan		25		17	
	ABPC+CEZ+CTX+SM+TC	1	4.0	1	5.0
	ABPC+SM+TC+NA	4	16.0	0	0.0
	SM+TC+NA	2	8.0	0	0.0
	SM+NA	0	0.0	1	5.0
	SM+TC	15	60.0	10	50.0
	TC	0	0.0	1	5.0
	KM	0	0.0	1	5.0
	SM	2	8.0	3	15.0
	susceptible	1	4.0	0	0.0

**Table 4 antibiotics-10-01541-t004:** Antimicrobial resistant genes in cefotaxime-resistant strains.

Serovar	Source	Strain	ST	Antimicrobial Resistance Profile	*bla* Gene	Other Antimicrobial Resistant Genes
Infantis						
	Human	BM-114	32	ABPC, CEZ, CTX, SM, KM, TC	CMY-2	*aac(6′)-Iaa*, *ant(3′′)-Ia*, *aph(3′)-Ia*, *qacEdelta1*, *sul1*, *tet*(A)
	Chicken	B-21	32	ABPC, CEZ, CTX, SM, TC	CMY-2	*aac(6′)-Iaa*, *ant(3′′)-Ia*, *dfrA14*, *qacEdelta1*, *sul1*, *tet*(A)
	Chicken	M-4	32	ABPC, CEZ, CTX, SM, TC	CMY-2	*aac(6′)-Iaa*, *ant(3′′)-Ia*, *qacEdelta1*, *sul1*, *tet*(A)
	Chicken	M-5	32	ABPC, CEZ, CTX, SM, TC	CMY-2	*aac(6′)-Iaa*, *ant(3′′)-Ia*, *dfrA14*, *qacEdelta1*, *sul1*, *tet*(A)
Blockley						
	Human	TK-117	52	ABPC, CEZ, CTX, SM, KM, TC, CP	CTX-M-15	*aac(6′)-Iaa*, *aph(3′′)-Ib*, *aph(3′)-Ia*, *aph(6)-Id*, *catA2*, *mph*(A), *tet*(A)
	Human	TK-120	52	ABPC, CEZ, CTX, SM, KM, TC, CP	CTX-M-15	*aac(6′)-Iaa*, *aph(3′′)-Ib*, *aph(3′)-Ia*, *aph(6)-Id*, *catA2*, *mph*(A), *tet*(A)
Minnesota						
	Human	TK-227	52	ABPC, CEZ, CTX, KM, TC	CMY-2	*aac(6′)-Iaa*, *aph(3′)-Ia*, *sul2*, *tet*(A)
	Human	TK-256	548	ABPC, CEZ, CTX, KM, TC, NA	CMY-2	*aac(6′)-Iaa*, *qnrB19*, *sul2*, *tet*(A)
Heidelberg						
	Human	TK-124	15	ABPC, CEZ, CTX, SM, GM, TC, NA	LAT-3	*aac(3)-VIa*, *aac(6′)-Iaa*, *ant(3′′)-Ia*, *fosA7*, *qacEdelta1*, *sul1*, *sul2*, *tet*(A)
	Human	TK-167	15	ABPC, CEZ, CTX, TC, NA	CMY-2	*aac(6′)-Iaa*, *fosA7*, *sul2, tet*(A)
Bareilly						
	Human	BM-153	203	ABPC, CEZ, CTX, SM, GM, TMP	CTX-M-55	*aac(3)-IId*, *aac(6′)-Iaa*, *ant(3′′)-Ia*, *dfrA14*, *fosA4*, *lnu*(F), *mph*(A), *qnrS13*
	Human	BM-226	203	ABPC, CEZ, CTX, SM, GM, TMP	CTX-M-55	*aac(3)-IId*, *aac(6′)-Iaa*, *ant(3′′)-Ia*, *dfrA14*, *fosA4*, *lnu*(F), *mph*(A), *qnrS13*
Manhattan						
	Human	BM-136	18	ABPC, CEZ, CTX, SM, TC	TEM-52B	*aac(6′)-Iaa*, *ant(3′′)-Ia*, *qacEdelta1*, *sul1*, *tet*(A)
	Chicken	K-1	18	ABPC, CEZ, CTX, SM, TC	TEM-52B	*aac(6′)-Iaa*, *ant(3′′)-Ia*, *qacEdelta1*, *sul1*, *tet*(A)
Goldcoast						
	Human	BM-274	358	ABPC, CEZ, CTX, GM, KM, TC, NA, CP	CTX-M-55, LAP-2, TEM-1	*ARR-3*, *aac(3)-IId*, *aac(6′)-Iaa*, *aph(3′)-Ia*, *aph(6)-Id*, *dfrA14*, *floR, qnrS13*, *sul2*, *sul3*, *tet*(A)
Untypable						
	Human	TK-272	32	ABPC, CEZ, CTX, SM, TC	CMY-2	*aac(6′)-Iaa, ant(3′′)-Ia, qacEdelta1, sul1, tet*(A)

ST: sequence type.

## Data Availability

Data are contained within the article.

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
