# Peer review of "Antimicrobial Resistance in Salmonella Isolated from Food Workers and Chicken Products in Japan"

_antibiotics, 2021, doi:10.3390/antibiotics10121541_

Round 1
Reviewer 1 Report
The authors are suggested to include minor editing to improve the manuscript as follows:
- The Material and Methods section must include a paragraph describing the statistical methodologies used in this work.
- Regarding the antimicrobial susceptibility tests (L 233 to L 237), it is recommended to include the commercial name, trademark, and city and country of origin of each of the antibiotics included, if possible.
- The paragraph under the subheading "Characterisation of Salmonella strains isolated from domestic chicken products" (L 211) could be improved by adding the methodology which was used to obtain and characterize the chicken-origin Salmonella isolates.
- The Conclusions section can be improved by including all the highlighted remarks
- The term "domestic chicken" (L 61, L 66, L 76, L 154, L 158, L 167, L 170, L 177, L 181, L 211, L 212 and L 260) widely used in this manuscript is not entirely correct, to my knowledge. I suggest using the terms "local chicken" or "local chicken products" instead of "domestic chicken" or "domestic chicken products", respectively.
Author Response
Response to Reviewer 1 Comments
We thank you very much for your kind and beneficial advice. Based on the comments and suggestions, we have revised the manuscript and our revisions are shown in red font.
We have also made editorial and consequential amendments.
Point 1: The materials and Methods section must include a paragraph describing the statistical methodologies used in the work.
Response 1: We prepared a new paragraph (4.6 Statistical analysis) (L289-L293) describing the statistical methods used in the study.
Point 2: Regarding the antimicrobial susceptibility tests (L233 to L237), it is recommended to include the commercial name, trademark, and city and country of origin of each of the antibiotics included, if possible.
Response 2: We do not have the information on the antibiotics because we purchased 96-well microtiter plates (Dry-plate ‘Eiken’) from the Eiken Chemical Co. We have added the name (Dry-plate ‘Eiken’) (L263-L264) in the respective paragraph.
Point 3: The paragraph under the subheading “Characterisation of Salmonella strains isolated from domestic chicken products”(L211) could be improved by adding the methodology which was used to obtain and characterize the chicken-origin Salmonella isolates.
Response 3: We have changed the subheading and the paragraph as follows “4.2 Local chicken products and Salmonella isolation” and “A total of 235 local chicken products (liver, breast, thigh, neck skin, and minced meat) were collected from 25 retail stores and 6 chicken processing plants between January 2018 and October 2021. The products were transported to the National Institute of Health Sciences under refrigeration. At the laboratory, each sample was tested within 24 h after arrival. Twenty-five grams of the sample was mixed in 225 mL of buffered peptone water (Oxoid Ltd, Hampshire, UK) and incubated at 37 °C for 18 h for pre-enrichment. After incubation, 0.1 and 1 mL of the culture was added to 10 mL of Rappaport–Vassiliadis broth (Oxoid) and 10 mL of Hajna tetrathionate broth (Eiken Chemical), respectively, and incubated at 42 °C for 20 h. After incubation, each culture was streaked onto two selective isolation agar plates: xylose-lysine-deoxycholate agar (Oxoid) and CHROMagarTM Salmonella (CHROMagar, Paris, France) and incubated at 37 °C for 24 h. For the selective isolation from minced meat, Mannitol-Lysine-Crystal-Violet-Brilliant-Green agar (Nissui Pharmaceutical, Tokyo, Japan) was used instead of xylose-lysine-deoxycholate agar. Putative Salmonella colonies were biochemically identified as mentioned above. One strain per sample was suspended in 20% glycerol and stored at −80 °C until serotyping and antimicrobial susceptibility testing.” (L232-L248).
Point 4: The Conclusions section can be improved by including all the highlighted remarks.
Response 4: We have changed the Conclusions to “Salmonella was isolated from 0.079% (583/740,635) of stool samples obtained from food workers in Japan. The strains were serotyped into 76 serovars and 22 untypeable Salmonella isolates. The top 14 serovars together accounted for more than 70% of the human-derived strains. Most of the top 14 serovars in human-derived strains are known to be prevalent in Japanese broiler and laying hen flocks. Of the 583 Salmonella strains, 12 (2.1%) were resistant to the TGC cefotaxime and harbored the same bla genes as those observed in local and imported chicken products. The characterization of Salmonella isolated from food workers showed that local and imported chicken products may represent the main sources of TGC-resistant Salmonella that infect food workers. Although the prevalence of TGC-resistant Salmonella in food workers is low, they may contaminate the food served to consumers. Therefore, monitoring the prevalence and characteristics of Salmonella in food workers is necessary for the risk management of Salmonella-associated food poisoning.” (L294-L307)
Point 5: The term “domestic chicken” (L61, L66, L76, L154, L158, L167, L170, L177, L181, L211, L212 and L280) widely used in this manuscript is not entirely correct, to my knowledge. I suggest using the terms “local chicken” or “local chicken products“, respectively.
Response 5: We have changed the term “domestic” to “local”.
We believe that the revised manuscript has been improved satisfactorily and will be accepted for publication in Antibiotics.
Reviewer 2 Report
Authors in this study determined the prevalence of Salmonella in food workers and characterize the isolates via serotyping and antimicrobial testing. they have used NGS platform to determine the antimicrobial resistance genes among the resistant strains. The outcome of this study are very much useful especially considering food workers as a reservoir for Salmonella. Information on food workers acquiring TGC-resistant salmonella is critical and limited studies are available specific on large sample collection. The quality of research in this study is good. Please see below comments that must be improved in the manuscript.
In abstract: re-correct characterize spelling and in the rest of the article.
In the introduction: I am surprised authors did not justify the outbreaks metrics on salmonella. please provide the global burden on the salmonella based on antimicrobial usage and their resistance. I recommend to provide more info on how did authors correlate Salmonella antimicrobial resistance from sample collected from workers and poultry. Provide what are the significance of antimicrobial resistance and their impact on human health.
In methods section:
- put a separate section on statistics and describe how did you compare.
- Provide clear and additional information on sample collection.
- L238-239: describe were the concentrations in 96 well plate were two fold diluted?
In discussion, 134-139- provide a relating/open statement on L151 (i.e. the link between workers and poultry isolates). It will be good if authors compare their results global studies and not specific to Japan such as the US, China, UK and other countries that will substantiate current findings and increases the value of this paper.
Author Response
We thank you very much for your kind and beneficial advice. Based on the comments and suggestions, we have revised the manuscript and our revisions are shown in red font.
We have also made editorial and consequential amendments.
Point 1: In abstract: re-correct characterize spelling and in the rest of the article.
Response 1: We have changed “characterise” to “characterize” throughout the manuscript.
Point 2: In the introduction: I am surprised authors did not justify the outbreaks metrics on Salmonella. Please provide the global burden on the Salmonella based on antimicrobial usage and their resistance. I recommend to provide more info on how did authors correlate from workers and poultry. Provide what are the significance of antimicrobial resistance and their impact on human health.
Response 2: We have added some sentences “Invasive non-typhoidal Salmonella is estimated to cause 3.4 million cases of infections annually, worldwide.” (L41-L42), “Therefore, antimicrobial-resistant Salmonella poses an important problem from the viewpoint of chemotherapy in humans.” (L48-L49), “TGCs are classified as “critically important” by the World Health Organization. In Japan, ceftiofur, one of TGCs, was approved in 1996 and has been used for chemotherapy in cattle and pigs with a bacterial infection. Moreover, the off-label use of ceftiofur in combination with in-ovo vaccination or vaccination of newly hatched chicks was performed in some hatcheries until the use was voluntarily abandoned in March 2012. The recent isolation of TGC-resistant Salmonella from chicken meat showed that TGC-resistant Salmonella might have still survived on hatcheries and chicken farms after the withdrawal of ceftiofur in hatcheries.” (L58-L65) and “To estimate the origin of the isolates, we also characterized Salmonella strains isolated from local chicken products by serotyping and antimicrobial susceptibility testing and compared the characteristics between human- and chicken-derived strains.” (L71-L73) in the introduction.
Point 3: In methods section, put a separate section on statistics and describe how did you compare.
Response 3: We have added a new paragraph (4.6 Statistical analysis) (L289-L293) in the section.
Point 4: In methods section, provide clear and additional information on sample collection.
Response 4: We have provided the additional information on sample collection. (L219-L228)
Point 5: In methods section; L238-239, describe were the concentrations in 96 well plate were two-fold diluted?
Response 5: The concentrations were two-fold diluted. We have changed “the broth microdilution method” to “the two-fold broth microdilution method”. (L263)
Point 6: In discussion, L134-139- provide a relating/open statement on L151 (i.e. the link between workers and poultry isolates). It will be good if authors compare their results global studies and not specific to Japan such as the US, China, UK and other countries that will substantiate current findings and increase the value of this paper.
Response 6: We have added “Xu et al. have suggested that Salmonella can cause human infections in both symptomatic and asymptomatic state because Salmonella strains isolated from symptomatic and asymptomatic individuals were genetically and phenotypically indistinguishable. Kariuki et al. reported a high level of relatedness between Salmonella strains isolated from symptomatic patients and asymptomatic carriers by phylogenetic analysis” (L146-L151) in the discussion section.
We believe that the revised manuscript has been improved satisfactorily and will be accepted for publication in Antibiotics.